# The Dynamics of Antimicrobial Resistance among Enterobacteriaceae Isolates in Russia: Results of the 2012–2018 INFORM and ATLAS International Program Studies

**DOI:** 10.3390/antibiotics11060790

**Published:** 2022-06-10

**Authors:** Roman Kozlov, Alexey Kuzmenkov

**Affiliations:** Smolensk State Medical Academy, 214019 Smolensk, Russia; alexey.kuzmenkov@eol-labs.com

**Keywords:** antimicrobial resistance, multidrug resistance, Enterobacteriaceae, ATLAS, INFORM, antibiotics, ESBL

## Abstract

*Background*: The increasing prevalence of multidrug-resistant Enterobacteriaceae limits the range of active antimicrobial agents, thus worsening clinical outcomes. The objective of this study was to identify the trends in antimicrobial resistance for Enterobacteriaceae in Russia using the databases for the International Network for Optimal Resistance Monitoring (INFORM) and Antimicrobial Testing Leadership and Surveillance (ATLAS) studies between 2012 and 2018. *Methods*: This subanalysis was performed for 3811 non-duplicate clinical isolates of Enterobacteriaceae to evaluate the in vitro activity of the main classes of antibiotics against relevant clinical isolates from hospitalized patients with complicated infections of different anatomical locations. *Results*: The lowest susceptibility was observed for colistin (0%), ampicillin (16.4%), and ampicillin/sulbactam (31.1%), whereas the best susceptibility was observed for all combinations containing avibactam (>96%). Among individual antimicrobials, doripenem (3.2%), tigecycline (1.6%), and meropenem (5.9%) exhibited the lowest resistance. Important trends included the decreasing resistance of Enterobacteriaceae to glycylcyclines and the increasing resistance to aminoglycosides and carbapenems. *K. pneumoniae* strains were most aggressive in terms of the percentage of strains having multidrug resistance (8.3–18.3%, depending on location) and the percentage of ESBL-positive strains (44.8–86.8%). *Conclusions*: The current patterns and trends of antimicrobial resistance in different bacterial species should be taken into consideration for timely updating of clinical guidelines and local treatment protocols to ensure effective antimicrobial therapy.

## 1. Introduction

Enterobacteriaceae are among the most common pathogens, causing a broad range of community-acquired and healthcare-associated infections and associated with substantial worsening of patients’ quality of life and an increased mortality [1,2,3]. Enterobacteriaceae are of particular importance from the perspective of antimicrobial resistance due to their high potential for acquiring and developing resistance mechanisms to different classes of antibiotics with fast vertical and horizontal transfer [4,5,6]. Infections caused by resistant Enterobacteriaceae strains are associated with a 2–4-fold increase in mortality compared with susceptible strains and are challenging for treatment due to the limited choice of active therapeutic agents [7,8,9].

The extensive growth of the resistance of Enterobacteriaceae to β-lactams is a matter of particular concern [2,10]. In 2018, the World Health Organization included carbapenem-resistant and third-generation cephalosporin-resistant Enterobacteriaceae in the global priority pathogens list (global PPL) as a pathogen of critical priority for the research and development of new and effective antibiotic treatments [11]. Careful monitoring of resistance rates could be obtained in large epidemiological studies. Updated epidemiological data on antibiotic resistance make it possible to adapt the existing treatment strategies and to set the direction for the research of new therapeutic options.

The Antimicrobial Testing Leadership and Surveillance (ATLAS) Program is being undertaken to evaluate the longitudinal in vitro activity of various antibiotic classes against significant clinical isolates [12,13] derived from hospitalized patients with complicated urinary tract infections (cUTI), complicated intra-abdominal infections (cIAI), complicated skin and soft tissue infections (cSSTI), lower respiratory tract infections (LRTI), as well as isolates recovered from blood specimens. In this paper, we report the results of cUTI, cIAI, cSSTI, blood, and LRTI from the ATLAS study for the surveillance period 2012–2018. In addition, the trends of antibiotic resistance in extended-spectrum β-lactamase (ESBL)-producing bacteria for different types of infections (community- or hospital-acquired) were also analyzed. This work was aimed at analyzing the trends of antimicrobial resistance of Enterobacteriaceae in Russia under the ATLAS program.

## 2. Results

### 2.1. Distribution of Isolates

Among the total of 3811 isolates of Enterobacteriaceae, *E. coli* (n = 1314, 34.5% of all isolates) and *K. pneumoniae* (n = 1246, 32.7% of all isolates) were the two most common species. The percentage of the following species was >3% of the total number of isolates: *Enterobacter cloacae* (n = 297, 7.8% of all isolates), *Proteus mirabilis* (n = 266, 6.7% of all isolates), and *Klebsiella oxytoca* (n = 130, 3.4% of all isolates). The percentage of the remaining 26 species was <15% of all isolates (ranging from 2.6% to 0.03% each). The general characteristics of the isolates are shown in Table 1, Table 2, Table 3 and Appendix A, and Figure 1 and Figure 2.

Most isolates were collected from patients aged 45 years and older (60.8%); the percentage of pediatric patients (<18 years old) was 15.1%. The male to female ratio in the study was approximately equal. Most isolates were collected from the lower respiratory tract (32.4%), with the fewest isolates from intra-abdominal (16.1%) culture sources. The predominant type of referring wards were the non-intensive care unit (non-ICU) departments of hospitals (79.2%).

Among the two most prevalent genera, *Klebsiella* spp. was predominantly isolated from LTRI (49% of all *Klebsiella* spp. isolates), whereas *Escherichia coli* was isolated from UTI (38.2% of all *Escherichia coli* isolates) followed by LRTI (16.6% of all *Escherichia coli* isolates). The *Enterobacter* spp. was isolated more often from SSTI (30.4% of all *Enterobacter* isolates); the *Proteus* spp. isolates from UTI (35.8%); the *Citrobacter* spp. isolates from SSTI (30.3%); the *Serratia* spp. isolates from LRTI (54.6%); and *Morganella* isolates from UTI (36.5%).

### 2.2. Antimicrobial Susceptibility and Resistance

Appendix A and Table 4 list the cumulative percentages of a total of 3811 Enterobacteriaceae isolates by minimal inhibitory concentration (MIC) calculated for each antibiotic, along with the antimicrobial susceptibility and patterns. Based on the distribution of the MIC values and a susceptibility rate above 90%, the most active antimicrobial agents tested against Enterobacteriaceae isolates were aztreonam/avibactam (resistance rate, 0.4%; MIC_50/90_ = 0.06/0.25 mg/L), ceftazidime/avibactam (resistance rate, 2.1%; MIC_50/90_ = 0.12/1 mg/L), ceftaroline/avibactam (resistance rate, 3.8%; MIC_50/90_ = 0.06/0.25 mg/L), doripenem (resistance rate, 3.2%; MIC_50/90_ = 0.06/0.5 mg/L), tigecycline (resistance rate, 1.6%; MIC_50/90_ = 0.5/2 mg/L), and meropenem (resistance rate, 5.9%; MIC_50/90_ = 0.06/0.25 mg/L). Antimicrobial resistance rates >50% were demonstrated for four antibiotics: ciprofloxacin (resistance rate, 52.3%; MIC_50/90_ = 1/8 mg/L), ceftaroline (resistance rate, 57.3%; MIC_50/90_ = 16/256 mg/L), ampicillin/sulbactam (resistance rate, 58.2%; MIC_50/90_ = 32/128 mg/L), and ampicillin (resistance rate, 80.8%; MIC_50/90_ = 64/64 mg/L).

Table 5 summarizes the cumulative rates of antimicrobial resistance for all isolates, as well as the two most common species (*Escherichia coli* and *Klebsiella pneumoniae*) to different classes of agents, as well as trends for different years. The rate of fluoroquinolone resistance among Enterobacteriaceae isolates was consistently high during the entire study period: the cumulative percentage for Enterobacteriaceae and *E. coli* was 40%. For *K. pneumoniae*, it was much higher: 65.7% (*p* < 0.001 compared to that for *E. coli*). The percentage of isolates resistant to combinations of penicillins with a β-lactamase inhibitor, monobactams, cephalosporins, carbapenems, and aminoglycosides significantly varied among the *Enterobacterales* species (*p* < 0.001 in each case). Polymyxins and glycylcyclines exhibited statistically significant differences in neither antimicrobial drug resistance trends nor the cumulative percentage of resistant isolates for the bacterial species. Aminoglycosides, carbapenems, and combinations of penicillins with a β-lactamase-inhibitor displayed a statistically significant trend towards an increasing percentage of resistant strains, which was more pronounced for *K. pneumoniae* than for *E. coli*.

### 2.3. Multidrug Resistance of Isolates

The rate of multidrug-resistant (MDR) strains was calculated using two types of criteria (Appendix A, by Hackel et al. [14] and by Castanheira et al. [15]); the results ranged significantly, but similar trends were observed. Significant heterogeneity in multidrug resistance parameters was observed for different infection sources over the studied period, which can be attributed to the amendments made to the study protocol. The rate of multidrug resistance tended to increase for all infection source locations except for the IAI, and reached 49.2% for UTI, 48% for SSTI, and 66.9% for LRTI by 2018 as estimated by Hackel (2016), or 24.6% for UTI, 17.6% for SSTI, and 31.4% for LRTI as estimated by Castanheira (2019).

When the rates of multidrug resistance were compared for two main bacterial species, *K. pneumoniae* isolates showed higher rates compared with *E. coli* isolates: 834/1246 (66.9%) vs. 370/1314 (28.2%) according to Hackel, and 176/1246 (14.1%) vs. 23/1314 (1.8%) according to Castanheira (*p* < 0.001 in both cases). Meanwhile, no multidrug-resistant *E. coli* strains were detected in 2014 or 2016 (according to Castanheira).

According to Hackel 2016, MDR isolates were found across all infection source locations and years. For *E. coli*, the MDR rates were comparable and did not differ significantly for different locations of infection source, whereas it was found for *K. pneumoniae* that the percentage of MDR isolates was lower in the cases of intra-abdominal infections (8.3%) compared with those of UTI (13.9%, *p* = 0.107) and LRTI (14%, *p* = 0.091), and was statistically significantly lower compared with that in the cases of SSTI (18.3%, *p* = 0.009).

Figure 3 and Figure 4 show the dynamics of multidrug-resistant strains presented individually for each species and separately for *E. coli* and *K. pneumoniae* depending on the location of the infection source.

### 2.4. ESBL Characterization of the Isolates

Overall, ESBL-producing organisms accounted for 1784 (46.8%) of all Enterobacteriaceae isolates, with 529 (40.3%) of *E. coli* isolates and 935 (75%) of *K. pneumonia* isolates. Figure 5 shows the dynamics of distribution for ESBL-producing organisms over the study years depending on the location of the infection source (the distribution is also presented in Appendix A). Among all 1784 ESBL-positive isolates, genetic testing confirmed molecular changes in 1603 isolates (89.9%). Pathologic modifications were most frequently detected in four genes: CTXM1 (in 81.1% ESBL + isolates), CTXM9 (9.3%), SHV (7.4%), and TEM (6.3%). Among all the Enterobacteriaceae isolates, an obvious trend towards a reduction in the percentage of ESBL-positive isolates was observed only for IAI, with a statistically insignificant rise in 2017–2018. For *E. coli* isolates, the percentage of ESBL-positive strains tended to increase over time for UTI (from 23.6% in 2012 to 41.3% in 2018 (*p* = 0.064)) and decrease for SSTI, from 60.9% in 2012 to 38.2% in 2018 (*p* = 0.112). The dynamics of the distribution of ESBL-producing organisms over years depending on the location of the infection source for *Klebsiella pneumoniae* shows that the percentage of ESBL-positive isolates was significantly increased for all localizations except for IAI, and in 2018 reached 82.8% for UTI, 86.8% for LRTI, and 83.3% for SSTI (Figure 5). Detailed data on changes in the dynamics of the rate of ESBL-positive isolates depending on the location of the infection source for different bacterial species are shown in Appendix A. Among all the isolates, the highest percentage of ESBL-positive strains was observed for LRTI (59.8%), and was statistically significantly higher compared with the cumulative parameter for IAI (35.8%), UTI (39.3%), and SSTI (45.4%) (*p* < 0.0001 for each case). A similar prevalence was also observed for *K. pneumoniae* (81.2% vs. 59% (*p* < 0.001), 66.5% (*p* < 0.001), and 77.3% (*p* = 0.239), respectively) and for *E. coli* (although being less pronounced for *E. coli*: 49.5% vs. 40.4% (*p* = 0.130), 33.3% (*p* = 0.897), and 44.7% (*p* = 0.007), respectively). Overall, the percentage of ESBL-positive strains was statistically significantly higher among *K. pneumoniae* isolates compared with *E. coli* isolates (75% vs. 40.3%, *p* < 0.001) and all other bacterial species.

## 3. Discussion

This study aimed to identify trends in the antimicrobial resistance of Enterobacteriaceae in Russia using the data from the International Network for Optimal Resistance Monitoring (INFORM) and ATLAS databases. Findings regarding in vitro antimicrobial resistance and resistant phenotypes among organisms collected in Russia in 2012–2018 are reported. The ATLAS program intends to assess the longitudinal in vitro activity of various antibiotic classes against relevant clinical isolates from inpatients having complicated infections of different anatomical locations. In this study, we identified the trends of antimicrobial resistance of Enterobacteriaceae in Russia using the data from the INFORM and ATLAS databases.

*Enterobacterales* are an important cause of serious infections [16,17]. Many bacteria belonging to this family are now becoming increasingly resistant to existing antibiotics [18,19,20]. It is a very threatening trend that undoubtedly requires surveillance and active measures to prevent the further spread of resistance in these important Gram-negative pathogens [21,22,23,24]. Antimicrobial resistance (AMR) is a major threat to global public health [25,26,27]. This is confirmed by an abrupt increase in the number of publications (up to more than 100 annually) focusing on various aspects of antibiotic resistance in *Enterobacterales* and searching for new treatment strategies. Infections caused by multidrug-resistant bacteria are associated with increased mortality, extended length of hospital stay, and higher costs of healthcare [1,9,28].

Due to the growing prevalence of multidrug-resistant Enterobacteriaceae, the efficacy of existing antibacterial agents decreases with time, while the discovery and approval of new antimicrobial agents lag behind the spread of AMR; therefore, efforts should be taken for timely updating of the existing guidelines to ensure the efficacy of antimicrobial treatment [29,30,31].

The results of previous studies showed that the prevalence of AMR among clinical strains of Enterobacterales in Russia is rather high [25]. Both our and earlier studies demonstrate that all Enterobacterales species, and *K. pneumoniae* in particular, are often resistant to modern cephalosporins. We can see that the levels of resistance for *K. pneumoniae* and *E. coli* were steady in 2012–2018, whereas the trend was increasing for the remaining Enterobacterales strains.

Resistance to carbapenems (meropenem, doripenem, imipenem, and ertapenem) was observed in 5.9%, 3.2%, 10.6%, and 8.0% of all Enterobacteriaceae isolates, respectively. The highest rate of carbapenem resistance was observed in *K. pneumoniae* isolates (19.1%). These data concur with the previous studies conducted in Russia. Although carbapenems remain active against most (82.0–90.0%) clinical strains of Enterobacteriaceae, it is important to emphasize the growing rate of carbapenem-resistant Enterobacteriaceae, up to 60% for strains isolated in 2018. Other studies have also reported a growing prevalence of carbapenem-resistant and carbapenemase-producing Enterobacteriaceae in Russia with a wide diversity in the types of carbapenemases, which is due to the features of local antibiotic treatment protocols in Russia and the fact that patients can purchase antibiotics without prescription in many regions. Combinations of aztreonam and ceftazidime with avibactam, a novel β-lactamase inhibitor (aztreonam/avibactam and ceftazidime/avibactam), exhibited the highest activity against Enterobacteriaceae; the susceptibility of bacteria to these antimicrobials was 98.8% and 98.0%, respectively. The high in vitro activity of combinations of aztreonam and ceftazidime with avibactam opens up a new avenue for the treatment of infections caused by carbapenemase-producing Enterobacteriaceae strains, but data on the predominant resistance mechanism and local susceptibility are important for appropriate use of these new therapeutic options [29,32].

Among non-β-lactam antibiotics, amikacin and tigecycline exhibited the broadest activity (88.7% and 94.8% of susceptible isolates, respectively). Susceptibility to colistin was observed in 86.3% of isolates. Because of the high rate of multiple resistance to the conventionally used non-β-lactam antibiotics belonging to aminoglycoside group, such as gentamicin (32.7–65.0%) and fluoroquinolones (65.0%), the wide use of these antibiotics (both as monotherapy or as a component of combination therapy) also cannot be recommended except for cases where bacterial susceptibility has been confirmed or if there are relevant and valid local data about low prevalence of resistant bacterial strains.

In the SMART study conducted between January 2017 and December 2017 by the Taiwan Centers for Disease Control [20], ceftazidime/avibactam, ertapenem, and colistin showed the highest efficacy against *E. coli* (99.9%, 98.0%, and 98.8% susceptibility, respectively). The least effective agents in this study were ciprofloxacin and levofloxacin (59.9% and 63.8% susceptibility, respectively). The study using the INFORM global surveillance program database was conducted in 26 medical laboratories in six Latin American countries in 2012–2015 [33] and showed that ceftazidime–avibactam, tigecycline, doripenem, imipenem, meropenem, and colistin were most effective against *E. coli* (99.9%, 99.9%, 99.5%, 99.2%, 99.4%, and 99.5% susceptibility, respectively). Cefepime, aztreonam, and levofloxacin were the least effective antibiotics in this study (67.7%, 67.7%, and 51.9% susceptibility, respectively). In the study conducted using the REPRISE database [34], the following antimicrobials showed the highest effectiveness against *E. coli*: ceftazidime–avibactam, tigecycline, imipenem, meropenem, and colistin (100.0%, 100.0%, 100.0%, 99.4%, and 97.2% susceptibility, respectively).

The study conducted between January 2013 and September 2014 using the data from the PACT database [35] showed that ceftolozane/tazobactam, meropenem, amikacin and colistin were the most effective antibiotics against *E. coli* in Western Europe (99.1%, 99.9%, 97.9%, and 99.5%, respectively). In Eastern Europe, the highest activity was also observed for the same agents (ceftolozane/tazobactam, meropenem, amikacin, and colistin: 96.1%, 99.7%, 95.8%, and 99.6% susceptibility, respectively). Ceftazidime exhibited the lowest effectiveness in this study (both in Western and Eastern Europe): 84.5% and 67.8% susceptibility, respectively. We have revealed in our study that ceftazidime/avibactam combination, tigecycline, doripenem, meropenem, and colistin were the most effective antibiotics against *E. coli* (98.8%, 94.8%, 95.6%, 92.5%, and 99.4% susceptibility, respectively). Ceftaroline and ciprofloxacin were the least effective agents.

Overall, the findings obtained in our study are consistent with the data reported in five previous studies. Ceftazidime/avibactam was found to be among the most effective antibiotics in our study. We revealed that 98.2% of Enterobacteriaceae isolates were susceptible to ceftazidime/avibactam combination (MIC_90_ = 0.5 mg/L). Furthermore, the surveillance data for ceftazidime/avibactam in this study show agreement with the findings from three previous INFORM studies, where Enterobacteriaceae isolates were collected in the medical centers in the Asia-Pacific region and Europe in 2012–2015 [33]; all of these isolates exhibited >99% susceptibility to ceftazidime/avibactam using the FDA breakpoint of <=8 mg/L. In our study, we also found that 79.4% of Enterobacteriaceae isolates were susceptible to ceftolozane/tazobactam combination. This result was consistent with the findings reported in another study focusing on monitoring the antimicrobial susceptibility in the Asia-Pacific region (72%), but lower than the data reported in the study conducted in 2013–2015 (89.2%).

Carbapenem-resistant Enterobacterales (CRE) are among the greatest threats related to antimicrobial resistance, which might have the most significant effect on human health [20,36]. The number of U.S. facilities where CRE has been found is increasing steadily and includes 4% of emergency hospitals and 18% of long-term acute-care centers [37]. A prominent feature of CRE infections is that they are associated with poor clinical outcomes. Furthermore, the mortality rate in patients with bloodstream infection caused by carbapenem-resistant *Klebsiella pneumoniae* is reported to be 40–50% [38]. Our study shows that the rate of resistance of *Klebsiella pneumoniae* strains to carbapenems is also approximately 20% and rapidly increases with time, up to 60% in strains collected in 2018. Interestingly, the SMART data led researchers to conclude that relebactam restores susceptibility to imipenem for most of the tested imipenem non-susceptible isolates of *K. pneumoniae* and *P. aeruginosa*, as well as some isolates of *Enterobacter* species non-susceptible to imipenem.

The high rate of antibiotic-resistant Enterobacteriaceae makes it very challenging to choose a proper effective treatment and increases patient mortality. In order to refine treatment regimens and choose timely and effective therapy, the current patterns and trends in antimicrobial resistance for different bacterial species should be taken into account. According to the findings reported in this study, ceftazidime/avibactam, aztreonam/avibactam, and colistin are the consistently effective options for treating infections caused by bacteria belonging to the family Enterobacteriaceae. Doripenem, meropenem, tigecycline, and amikacin also exhibited consistent in vitro activity against most isolates. In order to identify clinically significant changes in the resistance profiles among Enterobacteriaceae, regular local monitoring of antimicrobial activity must be performed, with a special focus placed on informing healthcare facilities about the appropriate use of antibiotics.

## 4. Methods

### 4.1. Study Design

This subanalysis was performed using the registry data obtained from the ATLAS and INFORM surveillance programs. The ATLAS Program was undertaken to evaluate the longitudinal in vitro activity of different antimicrobial agents including tigecycline, linezolid, ceftaroline, ceftaroline/avibactam, and ceftazidime/avibactam against relevant clinical isolates. The subanalysis included the data on 3811 non-duplicate clinical isolates of Enterobacteriaceae that were collected from different specimens from patients hospitalized with IAI, UTI, SSTI, or LRTI. These isolates were collected at 16 Russian study centers according to local laboratory procedures for isolating pathogens from patient specimens that participated in the INFORM and ATLAS programs between 2012 and 2018. Isolates were identified at each site and shipped to a central reference laboratory (International Health Management Associates, Inc., Schaumburg, IL, USA) for species confirmation using matrix-assisted laser desorption ionization–time of flight spectrometry (Bruker Biotyper MALDI-TOF, Bruker Daltonics, Billerica, MA, USA). The recorded demographic data included patients’ gender and age; isolate source; ward or area in the hospital the isolate originated from.

All the isolates were identified by study center number. The isolates were stored in tryptic soy broth supplemented with glycerol at −70 °C and shipped to International Health Management Associates, Inc. (IHMA; Schaumburg, IL, USA) to perform susceptibility testing. Only the isolates regarded as potential pathogens causing infection in patients were included in this study. In each patient, only the first isolate was tested per infectious episode. No ethical approval was necessary for this study as the isolates were collected for routine diagnostic testing.

### 4.2. Antimicrobial Susceptibility Testing

Antimicrobial susceptibility testing was performed with frozen broth microdilution panels manufactured by International Health Management Associates for isolates collected in 2015 or 2016, and TREK (TREK Diagnostic Systems, Thermo Fisher Scientific, Oakwood Village, OH, USA) for isolates collected in 2017 or 2018, against 25 antimicrobial agents according to the Clinical and Laboratory Standards Institute methodology (CLSI M7-09, M7-10, M7-11) [39]. The studied antibiotics (alone or in combination with a β-lactamase inhibitor) included aminoglycosides (amikacin, gentamicin), carbapenems (doripenem, meropenem, ertapenem, imipenem), cephalosporins (cefepime, ceftriaxone, ceftaroline, ceftazidime/avibactam, ceftaroline/avibactam, ceftolozane/tazobactam, ceftazidime, cefoperazone/sulbactam), glycylcyclines (tigecycline), polymyxins (colistin), penicillins (amoxicillin/clavulanate, ampicillin/sulbactam, ampicillin, piperacillin/tazobactam), monobactams (aztreonam, aztreonam/avibactam), quinolones (ciprofloxacin, levofloxacin), and sulfonamides (co-trimoxazole). However, the sets of antibiotics being tested varied for different years. The MICs were interpreted using both the CLSI and the European Committee on Antimicrobial Susceptibility Testing (EUCAST) breakpoints. The MICs were then interpreted according to the EUCAST breakpoints version 10.0 [40]. In order to assess the resistance dynamics, the analysis involved only the antimicrobials for which representative data for at least three consecutive years were available in the database. These antimicrobials included 25 agents from eight main classes. An isolate was considered resistant to a class of antibiotics if it exhibited resistance to at least one antimicrobial belonging to this class. Multidrug-resistant Enterobacteriaceae was defined using two different methods. According to the criteria proposed by Castanheira (2019), the MDR status was defined as any isolate non-susceptible when applying the CLSI breakpoints to any isolate non-susceptible to penicillins in combination with a β-lactamase-inhibitor, fluoroquinolones, aminoglycosides, glycosides, glycylcyclines, and polymyxins [15]. According to the criteria proposed by Hackel (2016), the MDR status was defined against three or more drug classes, including aminoglycosides (amikacin), β-lactam/β-lactamase inhibitor combinations (amoxicillin/clavulanate, piperacillin/tazobactam), monobactams (aztreonam), cephalosporins (cefepime, ceftazidime), carbapenems (doripenem, ertapenem, imipenem, meropenem), and fluoroquinolones (levofloxacin) [14].

### 4.3. Characterization of Phenotype Features

The CLSI ESBL-phenotypic criteria for epidemiological detection of ESBL-producing organisms was used for isolates and defined as the MIC value SBL-phenotypic criteria for ceftazidime, and/or aztreonam [39]. For additional information about the molecular ESBL status, isolates with high levels of phenotypic ESBL resistance were screened for the genes encoding TEM, SHV, CTX-M, VEB, PER, GES, plasmid-encoded AmpC, KPC, OXA-48-like, NDM, IMP, VIM, GIM, and SPM, using multiplex PCR assays as described previously [41].

### 4.4. Statistical Analysis

Descriptive statistics were provided as absolute frequencies or medians with interquartile range. The Mann–Whitney U-test, or Pearson’s χ^2^ test, or Fisher’s exact test and non-parametric Kruskal–Wallis test by rank and median multiple comparisons were used depending on the type of data processed. Analysis of variance (ANOVA) was employed to assess the intragroup dynamics of parameters. All the reported *p*-values were based on two-tailed tests of significance; *p*-values < 0.05 were regarded as statistically significant. The STATISTICA 7.0 software (StatSoft, Tulsa, OK, USA) and RStudio software version 1.0.136 (Free Software Foundation, Inc., Boston, MA, USA) with R packages version 3.3.1 (R Foundation for Statistical Computing, Vienna, Austria) was used for the analyses.

## 5. Conclusions

Among all of the currently used antimicrobial agents, combinations with avibactam remain the most promising as they are characterized by more than 95% susceptibility. Among single antibiotics, the fewest number of resistant strains and the best responses were observed for doripenem, tigecycline, and meropenem. Extremely low rates of susceptibility and high resistance to ampicillin/sulbactam, ampicillin, and colistin were detected for Enterobacteriaceae. The cumulative resistance rate of Enterobacteriaceae to glycylcyclines has been substantially declining since the early 2010s, but their resistance rate to aminoglycosides and carbapenems has been increasing over the past two years. *K. pneumoniae* isolates, especially those collected from the sources of lower respiratory tract infections and skin and skin structure infections, remain the most aggressive among all of the Enterobacteriaceae species both in terms of the rate of multidrug-resistant strains and the rate of ESBL-positive strains.

## Figures and Tables

**Figure 1 antibiotics-11-00790-f001:**
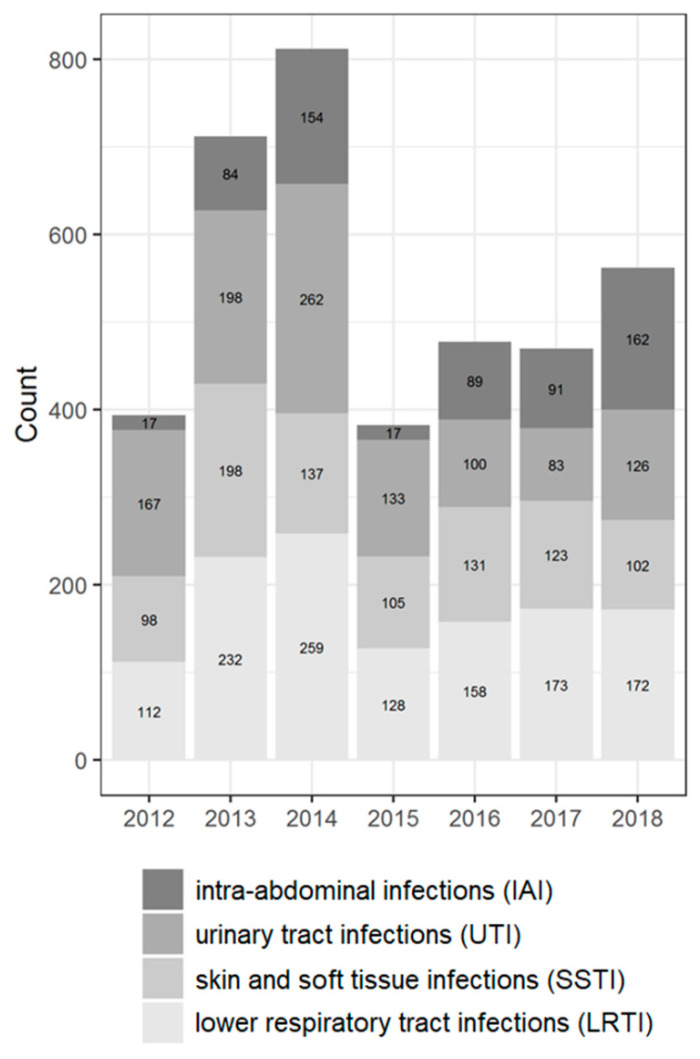
Distribution of the isolates according to source of infection by years (the absolute number and percentage of isolates for different years).

**Figure 2 antibiotics-11-00790-f002:**
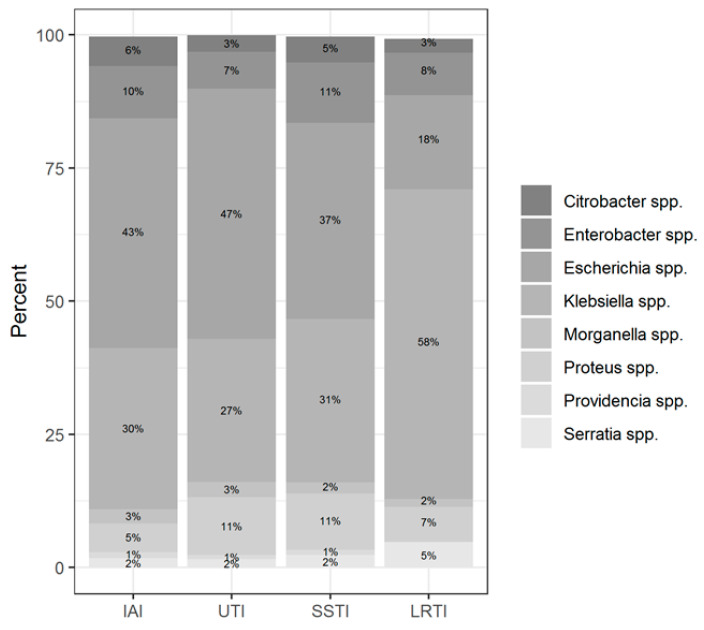
Distribution of the isolates according to infection localization.

**Figure 3 antibiotics-11-00790-f003:**
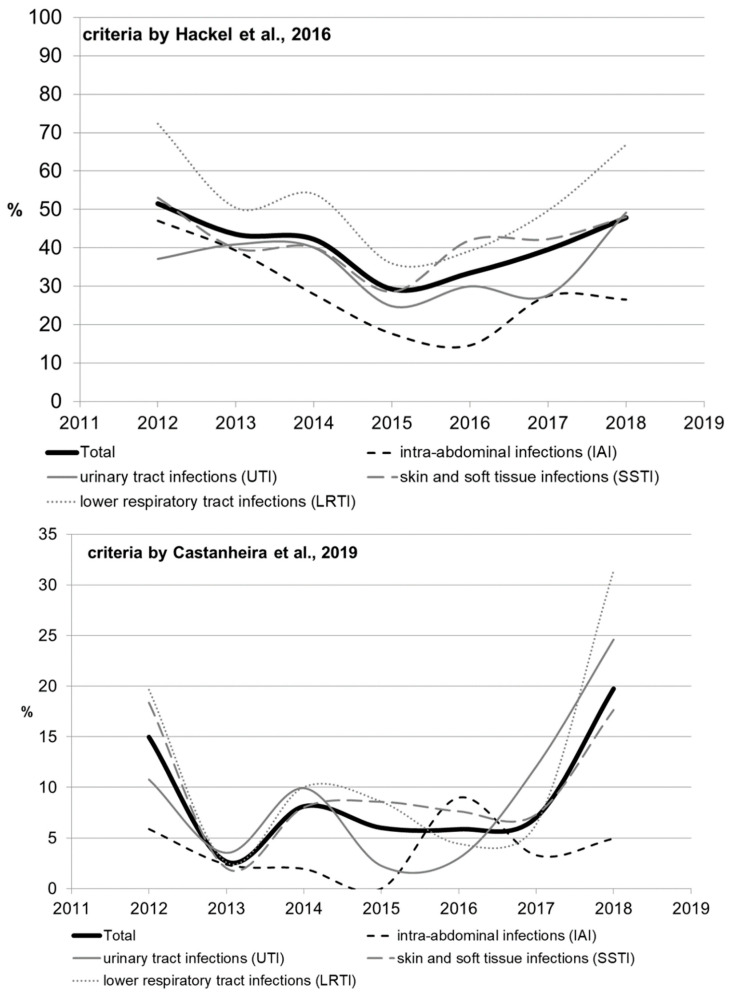
The dynamics of the percentage of multidrug-resistant strains among all *Enterobacterales* isolates depending on location of the infection source.

**Figure 4 antibiotics-11-00790-f004:**
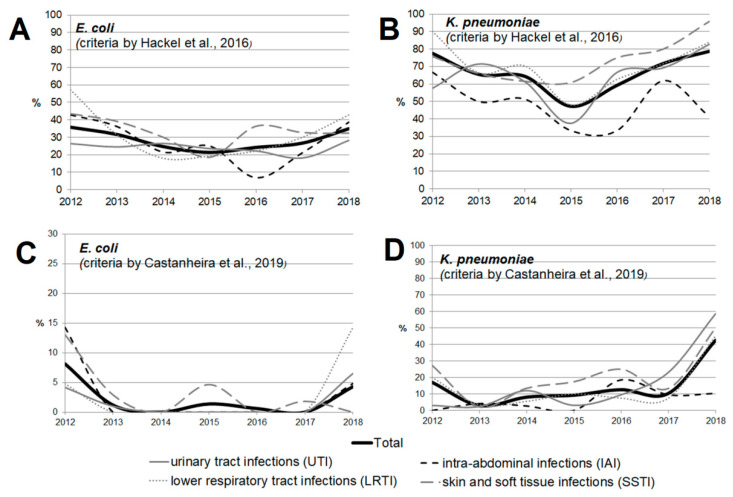
The dynamics of the percentage of multidrug-resistant strains among *E. coli* (**A**,**C**) and *K. pneumoniae* (**B**,**D**) isolates depending on location of the infection source according to the criteria proposed by Hackel et al. [14] (**A**,**B**) and Castanheira et al. [15] (**C**,**D**).

**Figure 5 antibiotics-11-00790-f005:**
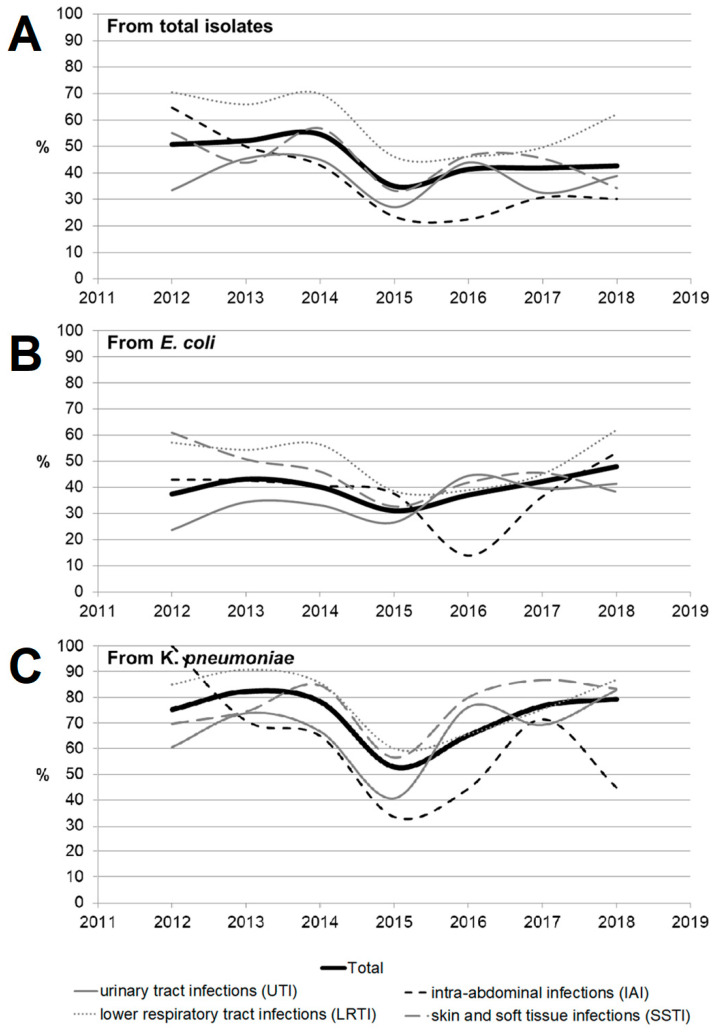
The dynamics of detection of ESBL-positive strains among Enterobacteriaceae isolates (**A**) depending on location of the infection source (as well as among the two most common species, *Escherichia coli* (**B**) and *Klebsiella pneumonia* (**C**)).

**Table 1 antibiotics-11-00790-t001:** The general characteristics of the isolates, N = 3811.

Parameter		Value
Patient gender:	Male	1969 [51.67%]
	Female	1832 [48.07%]
	Unknown	10 [0.26%]
Patient age, Me (IQR)		54 (31:65)
Age group:	0 to 2 years	242 [6.35%]
	3 to 12 years	285 [7.48%]
	13 to 18 years	55 [1.44%]
	19 to 44 years	898 [23.56%]
	45 to 64 years	1290 [33.85%]
	65 to 84 years	922 [24.19%]
	85 and over	105 [2.76%]
	Unknown	14 [0.37%]
Source	Intra-abdominal infection	614 [16.11%]
	Urinary tract infection	1069 [28.05%]
	Skin and soft tissue infection	894 [23.46%]
	Lower respiratory tract infection	1234 [32.38%]
Referring ward	Non-intensive care unit	3018 [79.19%]
	Intensive care unit	753 [19.76%]
	Unknown	40 [1.05%]

**Table 2 antibiotics-11-00790-t002:** The overall distribution of the isolates according to species.

Species	Number of Isolates (% of the Total Number)
*Escherichia coli*	1314 [34.48%]
*Klebsiella pneumoniae*	1246 [32.69%]
*Enterobacter cloacae*	297 [7.79%]
*Proteus mirabilis*	255 [6.69%]
*Klebsiella oxytoca*	130 [3.41%]
*Citrobacter freundii*	100 [2.62%]
*Serratia marcescens*	100 [2.62%]
*Morganella morganii*	85 [2.23%]
*Klebsiella aerogenes*	76 [1.99%]
*Proteus vulgaris*	56 [1.47%]
*Citrobacter braakii*	24 [0.63%]
*Enterobacter asburiae*	16 [0.42%]
*Providencia rettgeri*	15 [0.39%]
*Klebsiella variicola*	12 [0.31%]
Enterobacter, non-speciated	11 [0.29%]
*Proteus hauseri*	10 [0.26%]
*Citrobacter koseri*	9 [0.24%]
*Raoultella ornithinolytica*	9 [0.24%]
*Enterobacter kobei*	6 [0.16%]
*Serratia liquefaciens*	6 [0.16%]
*Citrobacter farmeri*	5 [0.13%]
*Enterobacter ludwigii*	5 [0.13%]
*Providencia alcalifaciens*	5 [0.13%]
*Providencia stuartii*	5 [0.13%]
*Citrobacter amalonaticus*	4 [0.1%]
*Hafnia alvei*	3 [0.08%]
*Proteus penneri*	3 [0.08%]
*Pluralibacter gergoviae*	1 [0.03%]
*Raoultella planticola*	1 [0.03%]
*Serratia odorifera*	1 [0.03%]
*Serratia ureilytica*	1 [0.03%]

**Table 3 antibiotics-11-00790-t003:** Distribution of isolates according to the infection source.

Species	Total	Intra-Abdominal Infections	Urinary Tract Infections	Skin and Soft Tissue Infections	Lower Respiratory Tract Infections
All Isolates	N = 3811	N = 614	N = 1069	N = 894	N = 1234
*Klebsiella* spp.	1464 [38.42%]	186 [30.29%]	287 [26.85%]	274 [30.65%]	717 [58.1%]
*Escherichia* spp.	1314 [34.48%]	265 [43.16%]	502 [46.96%]	329 [36.8%]	218 [17.67%]
*Enterobacter* spp.	335 [8.79%]	60 [9.77%]	74 [6.92%]	102 [11.41%]	99 [8.02%]
*Proteus* spp.	324 [8.5%]	33 [5.37%]	116 [10.85%]	94 [10.51%]	81 [6.56%]
*Citrobacter* spp.	142 [3.73%]	34 [5.54%]	33 [3.09%]	43 [4.81%]	32 [2.59%]
*Serratia* spp.	108 [2.83%]	11 [1.79%]	17 [1.59%]	21 [2.35%]	59 [4.78%]
*Morganella* spp.	85 [2.23%]	16 [2.61%]	31 [2.9%]	19 [2.13%]	19 [1.54%]
*Providencia* spp.	25 [0.66%]	7 [1.14%]	8 [0.75%]	9 [1.01%]	1 [0.08%]
*Raoultella* spp.	10 [0.26%]	1 [0.16%]	1 [0.09%]	2 [0.22%]	6 [0.49%]
*Hafnia* spp.	3 [0.08%]	1 [0.16%]	0 [0%]	0 [0%]	2 [0.16%]
*Pluralibacter* spp.	1 [0.03%]	0 [0%]	0 [0%]	1 [0.11%]	0 [0%]

**Table 4 antibiotics-11-00790-t004:** Cumulative percentages of Enterobacteriaceae isolates inhibited by different concentrations of antimicrobials. The MIC_50_ and MIC_90_ values are shown in light gray and dark gray, respectively. Yellow, orange, red, and dark red cells indicated antibiotics with resistance level more than 30%, 40%, 50%, and 80%, respectively.

	Concentrations, mg/L															Cumulative Rate		
Antimicrobial	≤0.015	0.03	0.06	0.12	0.25	0.5	1	2	4	8	16	32	64	128	≥256	Susceptible	Intermediate	Resistant
Aztreonam/avibactam	16.2	39.3	68.6	87.9	96.1	98.2	98.8	99.3	99.6	99.8	99.9	99.9	99.9	99.9	100	98.8	0.8	0.4
Ceftazidime/avibactam	3.2	9.9	29.7	55.4	74.2	88.9	95.4	97.2	97.9	98	98	98	98	98.1	100	98	0	2.1
Ceftaroline/avibactam	7.9	27.6	53.7	77.9	90.2	96.2	98.8	99.5	99.6	99.7	99.8	99.8	99.8	99.8	100	96.2	0	3.8
Doripenem	4	36.6	66.6	81.3	88.6	93	95.6	96.8	97.7	99.1	100					95.6	1.1	3.2
Tigecycline		0	1.7	15.3	41.8	68.4	85.2	94.8	98.5	99.8	100					94.8	3.7	1.6
Meropenem	8.8	42.3	77.9	87.8	90.6	91.3	92.5	94.1	95.6	96.2	98.3	100				92.5	1.6	5.9
Ertapenem	27.9	46.6	59.4	71	79.4	89.1	92	100								89.1	3	8
Amikacin					0.1	2.8	21.5	50.5	74.8	85.3	88.7	89.8	97.4	100		88.7	1.1	10.2
Imipenem		0	2	26.3	58.9	74	82.7	89.4	95.4	96.8	100					82.7	6.7	10.6
Ceftolozane/tazobactam	0.2	0.4	1.9	21.7	54.5	67.5	76.4	79.4	82.1	85.3	88.5	89.6	100			79.4	2.8	17.9
Piperacillin/tazobactam				0.2	5	10.1	22.9	47.2	60.7	67.1	75.2	80.1	83.5	89.9	100	75.2	8.3	16.5
Cefoperazone/sulbactam		0	1.6	22.6	29	39.2	43.6	48.2	55.2	62.1	68.7	82.8	87	100		68.7	14.1	17.3
Gentamicin				1.8	26.3	54.8	64.1	66.4	66.7	67.3	69	100				66.7	0.5	32.7
Ceftazidime	0.2	1.7	9	27.5	42.1	48.4	50.9	54.3	57.4	60.8	65.4	72	79.5	87.8	100	57.4	3.4	39.2
Aztreonam	6.4	11.7	29.1	42.6	47.2	49.1	50.4	51.5	54.1	57	61	68.1	77.4	89.9	100	54.1	2.9	43
Levofloxacin	0.1	16.4	31.9	38.7	47.4	53.1	56.9	59.6	63.5	85.8	100					53.1	3.8	43.1
Cefepime		0	0	45	47.7	49.4	51.1	52.9	55.3	59.1	63.8	94.6	100			52.9	6.2	40.9
Ceftriaxone			37.9	44.6	49.4	50.2	51.3	52	52.6	53.1	54.4	100				51.3	0.7	48
Co-trimoxazole							47.7	50.2	51.1	53	53.4	54.1	100			50.2	0	49.8
Ciprofloxacin				41.1	44.7	47.7	51.4	53.2	56.1	100						44.7	3	52.3
Ceftaroline	0.5	5	18.4	32.6	40.7	44.3	46.3	47.4	48.1	49.1	58.1	59.1	60.7	63	100	44.3	2	53.7
Amoxicillin/clavulanate					0	0.4	3.1	16.5	28.3	40.5	63.7	92.1	100			40.5	23.2	36.3
Ampicillin/sulbactam							2.5	8	19.6	31.2	41.8	57.5	68.3	100		31.1	10.7	58.2
Ampicillin						0.3	2	6.9	13.7	16.4	19.2	35.9	100			16.4	2.8	80.8
Colistin			0.2	2.5	29.5	64.9	85.2	86.3	86.8	90.7	100					0	86.3	13.8

**Table 5 antibiotics-11-00790-t005:** The dynamics of the resistance rates of *Enterobacterales*, *E. coli,* and *K. pneumoniae* isolates to different classes of antimicrobial agents.

Antimicrobial	Resistance Rate			Dynamics over Years 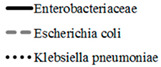
	Enterobacteriaceae	*E. coli*	*K. pneumoniae*	
Aminoglycosides	10.23%	2.97%	17.26%	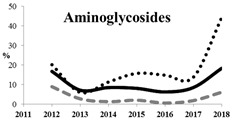
Carbapenems	11.6%	1.22%	19.1%	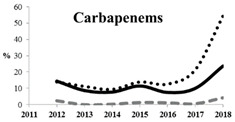
Cephalosporins	53.98%	45.21%	78.25%	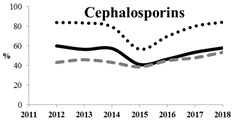
Glycylcyclines	1.55%	0%	0.4%	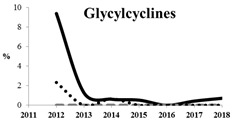
Polymyxins	9.76%	0.46%	0.8%	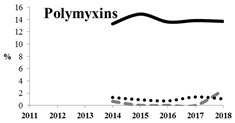
Monobactams	43.03%	33.33%	73.35%	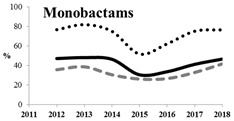
Penicillins in combination with a β-lactamase-inhibitor	38.65%	10.96%	48.64%	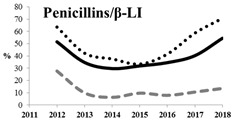
Quinolones	43.09%	42.16%	65.73%	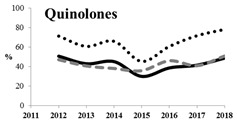

## Data Availability

Derived data supporting the findings of this study are available from the corresponding author on request.

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
