# Peer review of "The Dynamics of Antimicrobial Resistance among Enterobacteriaceae Isolates in Russia: Results of the 2012–2018 INFORM and ATLAS International Program Studies"

_antibiotics, 2022, doi:10.3390/antibiotics11060790_

Round 1

Reviewer 1 Report

This manuscript is well written and the subject is very important.

I proposed the title as follows: The Dynamics of Antimicrobial Resistance among Enterobacteriales Isolates in Russia: 2012-2018 because INFORM and ATLAS studies are national programs 

From the Abstract: I will suggest giving the meaning of INFORM or ATLAS

Author Response

I proposed the title as follows: The Dynamics of Antimicrobial Resistance among Enterobacteriales Isolates in Russia: 2012-2018 because INFORM and ATLAS studies are national programs 

We appreciate the suggested title change, however there is probably a typo or autocorrect in the proposed title as it is not entirely clear how the word "because" is used in context of period. We believe that the main message of the reviewer was to place an emphasis on the international volume of the program in the title, so we made changes based on this.

From the Abstract: I will suggest giving the meaning of INFORM or ATLAS

Program abbreviations have been added to the Abstract and Results section.

Reviewer 2 Report

  • Fig 1: One of the figure is enough. It is not nescessary both number and percentage.
  • Table 4: The first line should be “MIC values”.
  • Table 5, the last column: Enterobacteriaceae os Enterobacterales?
  • Methods 4.3: The reference is nescessary.
  • Which ESBL genes did you find?
  • It is sufficient to write clearly where the abbreviations appear first.

Author Response

Fig 1: One of the figure is enough. It is not nescessary both number and percentage.

Figure was changed.

Table 4: The first line should be “MIC values”.

This type of table does not imply the possibility of putting the MIC in the table header, since the MIC50 and MIC90 values can be found in any columns for each of the antibiotics. We have changed the table header and captions to make it clear to understand.

Table 5, the last column: Enterobacteriaceae os Enterobacterales?

Thank you for pointing out this inaccuracy. In our study, the main focus was on Enterobacteriaceae rather than Enterobacterales. However some of the cited articles describe the characteristics of Enterobacteriaceae in hole Enterobacterales only, which resulted in several typos during the translation of the article. All typos were corrected.

Methods 4.3: The reference is nescessary.

The reference was added

Which ESBL genes did you find?

Molecular testing was carried out on isolates that show high levels of phenotypic resistance. However, testing belonged to another part of the global survey; in our study testing was only used for additional clarifications of ESBL status and was not in main focus. Hole testing included genes as follows: ACC, ACTMIR, CMY1MOX, CMY11, FOX, IMP, KPC, SPM, SHV, TEM, CTXM1, CTXM2, CTXM825, CTXM9, VEB, PER, GES, DHA, OXA, NDM, VIM. Information about testing was corrected into Methods section. Main results of gene testing was added to Results section.

It is sufficient to write clearly where the abbreviations appear first.

Since the Methods section in the journal comes after the Results sections, many of the abbreviations have lost their transcription when originally mentioned. Thank you for bringing this to our attention - all abbreviations have been unified and deciphered in the order of appearance.

Reviewer 3 Report

 This manuscript aimed to analyze the trends of antimicrobial resistance of Enterobacterales in Russia under the ATLAS program. The paper is interesting. However, there are a few comments to improve the manuscript;

  1. Your main focus in this paper is Enterobacteriaceae. Please remove the data related to other pathogens.
  2. abbreviations need to be written in the full name for the first time in the manuscript.
  3. Add a table in the methods section to distribute your samples OR more table 1 to the study design section and merge numbers in figure 1 in the same table.
  4. Please list the screened 25 antibiotics used in this study and classify them according to the class.
  5. Provide more details on the methods, condition, media, etc for antibiotics screening with references.
  6. Please provide details on the isolation process for each pathogen.
  7. Why do you test the genotypic analysis only for genes encoding ESBLs?
  8. You need to provide correlation analysis between phenotypic and genotypic analysis with referral to correlation analysis performed in such papers https://www.mdpi.com/2079-6382/10/12/1450 .
  9. The figure on page 43 has no legend.
  10. In figure 4, add A, B, C, D and provide more details for each figure.
  11. Figure 5, same as figure 4.

Author Response

  1. Your main focus in this paper is Enterobacteriaceae. Please remove the data related to other pathogens.

Thank you for pointing out this inaccuracy. In our study, the main focus was on Enterobacteriaceae rather than Enterobacterales. However some of the cited articles describe the characteristics of Enterobacteriaceae only among hole Enterobacterale, which resulted in several typos during the translation and proofreading of the article. All typos were corrected.

  1. abbreviations need to be written in the full name for the first time in the manuscript.

Since the Methods section in the journal comes after the Results sections, many of the abbreviations have lost their transcription when originally mentioned. Thank you for bringing this to our attention - all abbreviations have been unified and deciphered in the order of appearance.

  1. Add a table in the methods section to distribute your samples OR more table 1 to the study design section and merge numbers in figure 1 in the same table.

Figure 1 has been modified but left in the current section based on your comments and comments from other reviewers.

  1. Please list the screened 25 antibiotics used in this study and classify them according to the class.

Full list of antibiotics with classes was added into Methods section.

  1. Provide more details on the methods, condition, media, etc for antibiotics screening with references.

Protocol information and related references have been added to the Methods section.

  1. Please provide details on the isolation process for each pathogen.

Protocol information and related references have been added to the Methods section.

  1. Why do you test the genotypic analysis only for genes encoding ESBLs?

In the hole INFORM and ATLAS group of studies and surveys, other beta-lactamases were analyzed on selected isolates for carbapenemases (class A, B, and D). Since they are not in the focus of this study, this why description restricted to the ESBL genes as to the main subject. Nevertheless. we considered it an important addition to expand the description of the ESBL-studied genes and also added brief information to the sections Methods and Results.

  1. You need to provide correlation analysis between phenotypic and genotypic analysis with referral to correlation analysis performed in such papers https://www.mdpi.com/2079-6382/10/12/1450 .

Although visualization variant of correlation analysis presented in the article you cited is not standard (and can only be reproduced in certain types of programs), it is undoubtedly of some interest to the reader due to its high visual informativeness. However, correlation analysis between genetic and phenotypic parameters in our study design can’t be performed, since the criterion for genotyping was the identification of the ESBL phenotype, which means that there was no comparison group without the ESBL phenotype. Only the presence of such a group may allow to perform statistical correlation or regression analysis.

  1. The figure on page 43 has no legend.

This figure (Figure 3) consists of two parts, the legend for both of them is located on the next page.

  1. In figure 4, add A, B, C, D and provide more details for each figure.

Figure 4 and its legends were modified.

  1. Figure 5, same as figure 4.

Figure 5 and its legends were modified.

Reviewer 4 Report

General Comments: This manuscript is focused on evaluation of the dynamics of antimicrobial resistance among Enterobacterales Isolates in Russia from two different databases in the period 2012-2018. Data are important and include relevant analysis of trends.

Some parts of Results sound as repetitions and it might be clearer if certain parts are included as supplementary materials. Material and Methods section is short. The procedure is not described in detail. The authors mentioned confirmation of ESBL strains by detection of specific genes (mentioned only in M&M section), but there are no results of this in the text. Why? E. coli represents a vast group of microorganisms and not all of them are of public health concern. Did the authors confirm pathogenic strains with some additional tests? This would provide valuable information for AMR. Discussion includes adequate information, but some parts are not convincing (see for details below). To make article easier to follow, these parts should be rewritten.

Specific comments:

Lines 39-41. Confusing sentence. Consider rewriting.

Figure 1. Is it necessary to include both graph (count and percentage), when these data are already present in the text. Maybe it could be part of supplementary materials.

Figures 3 and 4. Suggestion: It might be more appropriate to include it as supplementary materials.

Lines 66-72. Check grammar and style. It is not clear why authors sometimes use abbreviations and sometimes not. It should be uniform through the manuscript.

Lines 92-103. This paragraph does not convincingly explain the importance of surveillance of AMR. In addition, this is a "big group" of organisms. Their members have different significance for "global public health". Since, AMR indeed is of great importance; more precise information should be included here. Authors should consider rewriting this part.

Lines 113-125. How do authors explain "the growing rate of carbapenem-resistant Enterobacterales"? Check for grammar and style here too. Could this data be compared with other "global data"? The later paragraph (172-183) is better organised.

Lines 162-171. Do the previous INFORM studies included Family Enterobacteriaceae or Order Eneterbacterales?

Lines 213-214. What were the criteria for identifying "potential pathogens"?

Lines 218-223. Why were the "sets of antibiotics" different? It is not clear how MICs were interpreted.

Author Response

Some parts of Results sound as repetitions and it might be clearer if certain parts are included as supplementary materials. Material and Methods section is short. The procedure is not described in detail.

Thanks for the extended comment. The methods section has been expanded and supplemented with links. In the Results section, figures 1, 4, 5 and their captions have been changed.

The authors mentioned confirmation of ESBL strains by detection of specific genes (mentioned only in M&M section), but there are no results of this in the text. Why?

Molecular testing was carried out on isolates that show high levels of phenotypic resistance. However, testing belonged to another part of the global survey; in our study testing was only used for additional clarifications of ESBL status and was not in main focus. Hole testing included genes as follows: ACC, ACTMIR, CMY1MOX, CMY11, FOX, IMP, KPC, SPM, SHV, TEM, CTXM1, CTXM2, CTXM825, CTXM9, VEB, PER, GES, DHA, OXA, NDM, VIM. Information about testing was corrected into Methods section. Main results of gene testing was added to Results section.

  1. coli represents a vast group of microorganisms and not all of them are of public health concern. Did the authors confirm pathogenic strains with some additional tests? This would provide valuable information for AMR.

No strain typing was performed at IHMA for Sequence Type.  Isolates were deemed to be pathogens by the local laboratories and attending physicians (information added).

Discussion includes adequate information, but some parts are not convincing (see for details below). To make article easier to follow, these parts should be rewritten.

We have corrected the typos and individual comments you provide below in the text of the Discussion.

Specific comments:

Lines 39-41. Confusing sentence. Consider rewriting.

Sentence was rephrased.

Figure 1. Is it necessary to include both graph (count and percentage), when these data are already present in the text. Maybe it could be part of supplementary materials.

Figure 1 has been modified.

Figures 3 and 4. Suggestion: It might be more appropriate to include it as supplementary materials.

Figures have been modified but left in the current section based on your comments and comments from other reviewers.

Lines 66-72. Check grammar and style. It is not clear why authors sometimes use abbreviations and sometimes not. It should be uniform through the manuscript.

Since the Methods section in the journal comes after the Results sections, many of the abbreviations have lost their transcription when originally mentioned. Thank you for bringing this to our attention - all abbreviations have been unified and deciphered in the order of appearance. Sentence was rephrased also.

Lines 92-103. This paragraph does not convincingly explain the importance of surveillance of AMR. In addition, this is a "big group" of organisms. Their members have different significance for "global public health". Since, AMR indeed is of great importance; more precise information should be included here. Authors should consider rewriting this part.

We agree that Enterobacteriaceae event as part of Enterobacterales is a fairly large and heterogeneous group of microorganisms, each of which is characterized by its own characteristics of pathogenicity and importance. However, since 2019, when describing problems with antibiotic resistance or large epidemiological data, it has become the norm to combine such disparate data and this is done in most studies (629 of 675 articles containing Enterobacterales in the title were published for 2020-2022). We also present most of the trends for the whole group so that other researchers can use our data to compare with their local data. We expanded this section in Discussion.

Lines 113-125. How do authors explain "the growing rate of carbapenem-resistant Enterobacterales"? Check for grammar and style here too. Could this data be compared with other "global data"? The later paragraph (172-183) is better organised.

The text of this section has been changed

Lines 162-171. Do the previous INFORM studies included Family Enterobacteriaceae or Order Eneterbacterales?

It was both in different studies. We have adjusted mentions to match specific references.

Lines 213-214. What were the criteria for identifying "potential pathogens"?

No strain typing was performed at IHMA for Sequence Type.  Isolates were deemed to be pathogens by the local laboratories and attending physicians.

Lines 218-223. Why were the "sets of antibiotics" different? It is not clear how MICs were interpreted.

We have significantly modified this section to improve perception.

Round 2

Reviewer 3 Report

Thank you for addressing the comments, However, there is still few grammatical error that will need correction

Author Response

Thank you for addressing the comments, However, there is still few grammatical error that will need correction

Thank you for your feedback. We professionally proofread the text of the manuscript and corrected grammatical and stylistic errors throughout the text. All changes are made in review mode.

Reviewer 4 Report

General comments.

The article is approved and the majority of the comments were addressed. However, several questions remains to be checked (details are below).

After evaluating these comments, I recommend it for publishing in your Journal.

Page 5. Lines 94-96. Check for grammar, style and typing errors. Check also the same error in description of Table 4, and everywhere in the text after the word Enterobacteriaceae (missing space between words!)

Page 8. Lines 5-6

Page 16. Lines 216-217.  "Local laboratory procedure for isolating pathogens" should be something that is comparable, some standard method. The phrase "standard method" accompanied with proper reference would be more suitable here.

Page 17. Lines 261-264.  "Published multiplex PCR assays" should include relevant reference.

Author Response

The article is approved and the majority of the comments were addressed. However, several questions remains to be checked (details are below). After evaluating these comments, I recommend it for publishing in your Journal.

Thank you for your feedback. We professionally proofread the text of the manuscript and corrected grammatical and stylistic errors throughout the text. All changes are made in review mode.

Page 5. Lines 94-96. Check for grammar, style and typing errors. Check also the same error in description of Table 4, and everywhere in the text after the word Enterobacteriaceae (missing space between words!)

Corrections were made

Page 8. Lines 5-6

Corrections were made

Page 16. Lines 216-217.  "Local laboratory procedure for isolating pathogens" should be something that is comparable, some standard method. The phrase "standard method" accompanied with proper reference would be more suitable here.

Unfortunately, the local laboratory protocols that were used in the study cannot be cited since they refer to internal standards in Russian language and due to the absence of standard data for citation. However, we have added an extended phrase that is standard in all research on  ATLAS and INFORM study, which contains a description of the protocol actions.

Page 17. Lines 261-264.  "Published multiplex PCR assays" should include relevant reference.

Sentence was corrected and reference were added.